# *QIL1* mutation causes MICOS disassembly and early onset fatal mitochondrial encephalopathy with liver disease

Virginia Guarani[1†], Claude Jardel[2,3,4†], Dominique Chrétien[5], Anne Lombès[2,3,4], Paule Bénit[5], Clémence Labasse[6], Emmanuelle Lacène[6], Agnès Bourillon[7], Apolline Imbard[7], Jean-François Benoist[7], Imen Dorboz[5], Mylène Gilleron[2,3,4], Eric S Goetzman[8,9,10], Pauline Gaignard[11], Abdelhamid Slama[11], Monique Elmaleh-Bergès[12], Norma B Romero[6], Pierre Rustin[5], Hélène Ogier de Baulny[13], Joao A Paulo[1], J Wade Harper[1*], Manuel Schiff[5,13*]

[1]Department of Cell Biology, Harvard Medical School, Boston, United States; [2]Inserm U1016, Institut Cochin, CNRS UMR 8104, Paris, France; [3]Department of Biochemistry, APHP, GHU Pitié-Salpêtrière, Paris, France; [4]Université Paris-Descartes, Paris, France; [5]UMR1141, PROTECT, INSERM, Université Paris-Diderot, Sorbonne Paris Cité, Paris, France; [6]Neuromuscular morphology unit, Institut de Myologie, GHU Pitié-Salpêtrière, APHP, Paris, France; [7]Department of Biochemistry, Hôpital Robert Debré, APHP, Paris, France; [8]Department of Pediatrics, University of Pittsburgh School of Medicine, Pittsburgh, United States; [9]University of Pittsburgh, Pittsburgh, United States; [10]Children's Hospital of Pittsburgh of UPMC, Pittsburgh, United States; [11]Department of Biochemistry, Hôpital Bicêtre, APHP, Paris, France; [12]Department of Radiology, Hôpital Robert Debré, APHP, Paris, France; [13]Reference Center for Inborn Errors of Metabolism, Robert Debré University Hospital, APHP, Paris, France

*For correspondence:
wade_harper@hms.harvard.edu
(JWH); manuel.schiff@aphp.fr (MS)

†These authors contributed equally to this work

**Abstract** Previously, we identified QIL1 as a subunit of mitochondrial contact site (MICOS) complex and demonstrated a role for QIL1 in MICOS assembly, mitochondrial respiration, and cristae formation critical for mitochondrial architecture (*Guarani et al., 2015*). Here, we identify *QIL1* null alleles in two siblings displaying multiple clinical symptoms of early-onset fatal mitochondrial encephalopathy with liver disease, including defects in respiratory chain function in patient muscle. QIL1 absence in patients' fibroblasts was associated with MICOS disassembly, abnormal cristae, mild cytochrome *c* oxidase defect, and sensitivity to glucose withdrawal. QIL1 expression rescued cristae defects, and promoted re-accumulation of MICOS subunits to facilitate MICOS assembly. MICOS assembly and cristae morphology were not efficiently rescued by over-expression of other MICOS subunits in patient fibroblasts. Taken together, these data provide the first evidence of altered MICOS assembly linked with a human mitochondrial disease and confirm a central role for QIL1 in stable MICOS complex formation.

## Introduction

A diverse collection of mitochondrial diseases have been linked to mutations that affect either mitochondrial or nuclear genomes, and alter respiratory chain function, mitochondrial fission/fusion cycles, or mitochondrial quality control (*Vafai and Mootha, 2012*). A central feature of mitochondrial disease is the pleotropic nature of phenotypes, with some diseases eliciting highly tissue-specific

effects and others affecting multiple organ systems. The basis of this pleiotropy is poorly understood, but could reflect differences in mitochondrial structure and concomitant links with respiratory chain activity within individual tissues (*Vafai and Mootha, 2012*).

Proper organization of the respiratory chain and other mitochondrial protein complexes within the mitochondrial inner membrane (MIM) relies on the formation of MIM invaginations referred to as cristae. These laminar structures with highly curved distal membrane arrangement are formed through the mitochondrial contact site (MICOS) complex, a cristae junction scaffold that is thought to both maintain close proximity of cristae membranes near the 'boundary' MIM as well as form connections between the cristae junction and the mitochondrial outer membrane (MOM) (*Pfanner et al., 2014*; *van der Laan et al., 2016*). MICOS is composed of several subunits located on the MIM and mitochondrial intermembrane space, including MIC25, MIC60, MIC10, MIC26, MIC27, MIC19, and this core complex has been shown to interact with SAM50, MTX1, and MTX2 on the MOM, presumably helping to position cristae structures in proximity to the MOM. Our recent proteomic analysis of the MICOS complex identified a previously unstudied single-pass transmembrane protein – C19ORF70/QIL1 – as a component of MICOS localized in the MIM (*Guarani et al., 2015*). Depletion of QIL1 by RNAi in tissue culture cell lines resulted in: (1) loss of cristae junction structures accompanied by the formation of MIMs with characteristic 'swirl' structures, (2) MICOS complex disassembly, (3) loss of MIC10, MIC26 and MIC27 protein levels, (4) failure to assemble into a MIC60-MIC19-MIC25 sub-complex, and (5) decreased mitochondrial respiration (*Guarani et al., 2015*). RNAi of the *Drosophila* QIL1 ortholog in muscle and nervous tissue lead to a similar mitochondrial phenotype in vivo (*Guarani et al., 2015*). Thus, QIL1 is required for assembly and maintenance of the MICOS complex and mitochondrial architecture. This mechanism of MICOS assembly and the role of QIL1 have recently been verified (*Zerbes et al., 2016*; *Anand et al., 2016*).

Building upon our recent discovery and mechanistic analysis of QIL1, we now report recessive non-functional alleles of *QIL1* in two sibling patients with early onset fatal mitochondrial encephalopathy and recurrent liver disease. Molecular and cell biological studies revealed dramatic defects in mitochondrial organization in fibroblasts and muscle tissue derived from these patients that are associated with disassembly of the MICOS complex. MICOS assembly and mitochondrial architecture phenotypes in patient fibroblasts could be fully rescued by re-introduction of *QIL1*, but not other MICOS subunits, consistent with *QIL1* mutation underlying the phenotypes observed. These data reveal a critical role for the MICOS core complex in mitochondrial health and disease.

## Results

### Clinical presentation of patients deficient in QIL1

This study identified two patients from non-consanguineous parents with defects in the QIL1 gene. Both patients (patient 1, female and patient 2, male) were born after an uneventful pregnancy with normal birth parameters, although patient 1 had a smaller than the average head circumference (10th percentile). At 16 hr of life, patient 1 was admitted to the neonatal intensive care unit, displaying an array of abnormalities (*Figure 1A–C*, *Supplementary file 1A*), including hypothermia, lactic acidosis, hypoglycemia, and signs of liver failure. Plasma amino acids showed elevation of tyrosine and methionine concentration and urine organic acids revealed the presence of 3-methylglutaconic acid associated with lactic acid and Krebs cycle intermediates (malate and fumarate). After 48 hr of symptomatic treatment, including a protein and lipid-free infusion of glucose (13.5 g/kg/day) with electrolytes, the child fully recovered both clinically and biologically. During the next 6 months, neurodevelopment and growth were normal except for the incidence of acquired microcephaly from the age of 4 months (-2 SD at 4 months of age). At the age of 6 months, in the course of a benign febrile upper respiratory infection, she exhibited neurological deterioration with hypotonia, hyperlactacidemia with high lactate to pyruvate ratio, and evidence of mild liver disease without overt liver failure (*Supplementary file 1A*). Liver ultrasound disclosed hyperechogenicity of the liver with two hypoechogenic nodules in the VI and VII segments. Heart ultrasound disclosed mild hypertrophy with normal heart function and brain MRI showed cerebellar atrophy with optic atrophy (*Figure 1A–C*) and a moderate lactate peak on brain MR spectroscopy (not shown). Plasma amino acids showed elevation of methionine consistent with liver disease and urine organic acids showed persistent urinary excretion of 3-methylglutaconic acid (*Supplementary file 1A*). After three days of symptomatic treatment,

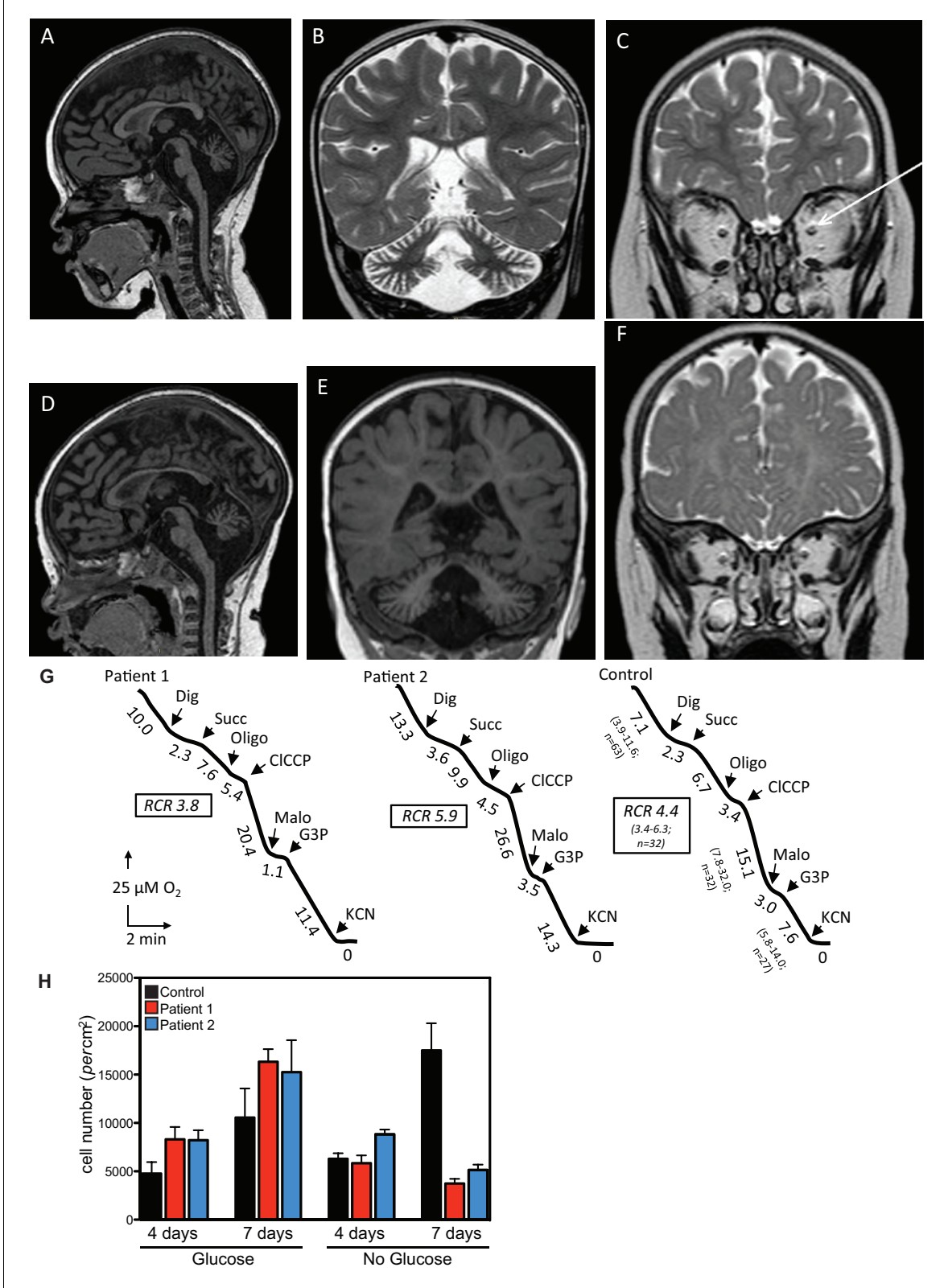

**Figure 1.** Clinical presentation of patients deficient in *QIL1*. (A–C) Brain MRI of patients 1 (A–C, 2 years of age) and 2 (D–F, 1 year of age). T1-weighted sagittal views show cerebellar atrophy of the vermis and the brainstem (A,D). T2-weighted (B, patient 1) and FLAIR (E, patient 2) coronal views show cerebellar hemisphere atrophy. T2-weighted coronal views (C,F) show optic atrophy (arrow, patient 1). (G) Oxygen consumption by patients' and control skin fibroblasts was measured as reported elsewhere (*El-Khoury et al., 2013*; *Rustin et al., 1994*). Cellular respiration was first measured using intact

*Figure 1 continued on next page*

*Figure 1 continued*

fibroblasts. It was essentially abolished upon addition of a limited amount of digitonin (Dig) which caused the leakage of endogenous respiratory substrates in the assay medium. Oxygen consumption resumed upon subsequent addition of 10 mM succinate (Succ) to the digitonin-permeabilized fibroblast. This oxidation decreased in the presence of oligomycin (oligo) (a mitochondrial ATPase inhibitor) while adding an uncoupling agent (*m*-ClCCP) allowed a maximal rate of oxidation and to calculate a respiratory chain control (RCR) value (rate in the presence of uncoupler *versus* rate in the presence of the ATPase inhibitor). Malonate (Malo) (a specific succinate dehydrogenase inhibitor) addition essentially abolished oxygen uptake linked to succinate oxidation. Adding glycerol-3 phosphate (G3P) allowed the oxygen uptake to resume, this latter being fully inhibited by the addition of cyanide (KCN). The values along the traces are nmol/min/mg protein. (H) Control (black bars) or patient (blue or red bars) fibroblasts were cultured in the presence and absence of glucose for 7 days and cell number determined after 4 and 7 days (n = 3).

there was a partial improvement of neurological functions, but persistence of poor eye contact and truncal hypotonia, disappearance of biological abnormalities except for mild hyperlactacidemia (2.5 to 3.5 mM). Bilateral and severe sensorineural deafness was confirmed by auditory evoked potentials. Severe bilateral neurovisual impairment was confirmed by evoked visual potentials, which showed barely detectable responses. Eye fundoscopy and elecroretinogram was normal. The child died before the age of 3 after slow progression of her neurological disease.

Patient 2, as with his sister, exhibited transient clinical and biological deterioration with hypothermia, lactic acidosis, hypoglycemia and liver dysfunction (*Supplementary file 1A*) within the first few hours after birth. Comparable biochemical abnormalities were observed with high plasma methionine and tyrosine and 3-methylglutaconic aciduria (*Supplementary file 1A*). Heart ultrasound was normal. At 6 months of age, investigations disclosed developmental delay, poor eye contact contrasting with normal fundoscopy, electroretinogram and visual evoked potentials, liver disease with mild elevation of liver enzymes, hyperlactacidemia and persistent urinary excretion of 3-methylglutaconic acid (*Supplementary file 1A*). At one year of age, he exhibited global hypotonia, delayed developmental milestones, normal growth for height, but weight was at −1.5 SD (0 SD at birth), acquired microcephaly (0 SD at birth, −2.5 SD at 1 year of age) and persistent elevation of liver enzymes (*Supplementary file 1A*) with ultrasound evidence of hyperechogenic nodular lesions in segments II and III. Acoustic otoemissions showed normal responses. As with patient 1, brain MRI showed cerebellar and optic atrophy (*Figure 1D–F*) with a moderate lactate peak on MR spectroscopy (not shown). Patient 2 died at the age of 20 months in the course of a febrile infection.

Taken together, these clinical findings are suggestive of neonatal onset mitochondrial encephalopathy with recurrent bouts of liver disease.

## Analysis of respiratory chain function in patient skeletal muscle and fibroblasts

Given the clinical findings, we examined respiratory chain function in skeletal muscle tissue and in cultured skin fibroblasts. Fibroblasts from both patients displayed mild complex IV (cytochrome *c* oxidase) deficiency when normalized to citrate synthase activity, but apparently normal overall respiration (*Figure 1G*, *Supplementary file 1B*). Compared with control fibroblasts, patient fibroblasts displayed reduced cell number 7 days after glucose withdrawal (*Figure 1H*), a characteristic often observed in mitochondrial disease and indicative of a limited capacity for mitochondria to respond to the metabolic challenge resulting from glycolysis limitation (*Lee et al., 2014*; *Robinson et al., 1992*; *van den Heuvel et al., 2004*). Finally, mitochondrial respiratory chain dysfunction was shown in the skeletal muscle biopsy of patient 1, which presented with deficiencies across all respiratory chain complexes (*Supplementary file 1C*). Differences in respiratory chain function in fibroblasts and muscle tissue may reflect adaptation of fibroblasts to in vitro culture conditions and/or tissue specific phenotypes. Sequence and quantification of mitochondrial DNA extracted from skeletal muscle were normal (Patient 1: 4500 copies/cell (n = 3); Control range: 1700–6000 copies/cell [15 infants]).

## Identification of mutations in *QIL1*

In order to search for mutations present in these patients, whole exome sequencing was performed. We identified a homozygous mutation (c.30-1G>A) in *C19orf70* (*QIL1*) in both patients (*Figure 2A*), and this mutation was confirmed by Sanger sequencing of PCR products from patient DNA extracted from fibroblasts. Parents were found to be heterozygous at this locus (*Figure 2A,B*),

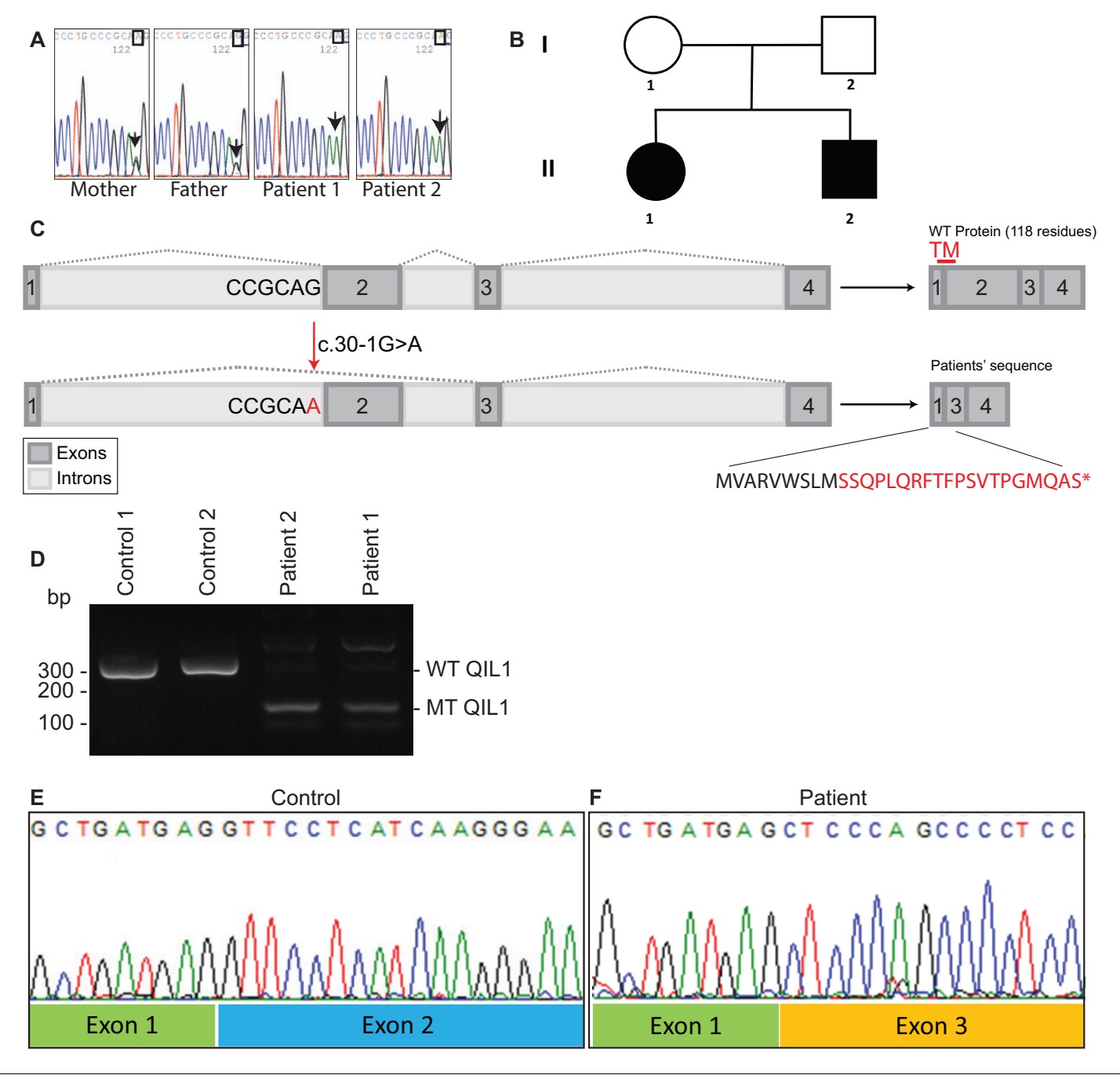

**Figure 2.** Identification of mutations in *QIL1*. (**A**) Chromatogram depicting a homozygous mutation (c.30-1G>A) in *C19orf70 (QIL1)*. (**B**) Pedigree showing autosomal recessive transmission. (**C**) Schematic illustration of the location of the mutation within the QIL1 transcript and the predicted consequences on splicing and the respective coding sequences. TM (red), indicates the position of a transmembrane domain in QIL1. (**D**) Ethidium bromide stained agarose gel of RT–PCR *QIL1* products from controls and patients 1 and 2. (**E–F**) Sequence analysis of control (**E**) and patient 1 (**F**) QIL1 cDNAs.

consistent with autosomal recessive transmission. Consistent with a G>A mutation in the 5' splice-donor sequence in exon 2, the QIL1 mRNA contained exon 1 fused to exon 3 (*Figure 2C–F*), which produced a frame-shift at Ser10 in the annotated QIL1 cDNA (NM_205767.2) leading directly into an alternative reading frame encoded by exon 3 and a premature stop codon (*Figure 2C*). Given that

the encoded putative protein contains only 9 residues from QIL1 and lacks the transmembrane domain primarily derived from exon 2, we conclude that the encoded protein corresponds to a functionally null allele (*Figure 2C*).

## Patient fibroblasts lacking QIL1 display defects in MICOS assembly and cristae structure

We next examined QIL1 expression and the abundance of MICOS complex proteins in control and patient-derived fibroblasts. Consistent with production of an altered QIL1 protein, we failed to detect a band at ~11 kDa corresponding to QIL1 in patient fibroblasts, but this protein was abundant in control fibroblasts based on immunoblotting of whole-cell extracts (*Figure 3A*). Previously we demonstrated that depletion of QIL1 leads to a reduction in the abundance of several MICOS complex subunits (*Guarani et al., 2015*). We therefore probed immunoblots with antibodies against six additional MICOS subunits. The abundance of all six proteins was reduced in both patient cell lines, with MIC10, MIC26 and MIC27 being affected to the largest extent (*Figure 3A,B*). Importantly, while QIL1 mRNA was greatly reduced in both patient cell lines relative to controls, the levels of mRNA for MIC10, MIC26, and MIC27 were largely unaffected in patient cell lines (*Figure 3C*). Thus, loss of MICOS subunits appears to occur through post-transcriptional mechanisms.

Since RNAi-induced QIL1 depletion had been shown to induce mitochondrial cristae abnormalities (*Guarani et al., 2015*), electron microscopy studies were performed both in patient fibroblasts and in patient 1 skeletal muscle biopsy. We observed severe cristae morphology abnormalities in the vast majority of mitochondria in fibroblasts, with >92% of mitochondria displaying loss of cristae structures and the appearance of characteristic 'swirl' structures similar to those previously seen upon depletion of MICOS subunits (*Figure 3D*) (*Alkhaja et al., 2012*; *Guarani et al., 2015*; *John et al., 2005*; *Weber et al., 2013*). While the majority of mitochondria in control fibroblasts were tubular in structure, mitochondria in patient fibroblasts were enlarged and swollen, round or elongated, consistent with a dramatic loss in architecture. Similarly, mitochondria in skeletal muscle from patient 1 were also swollen and filled with inner membranes displaying 'swirl' structures (*Figure 3E*).

## Introduction of QIL1 into patient fibroblasts rescues mitochondrial cristae morphology

In order to confirm the causative role of QIL1 loss in mitochondrial defects in patient fibroblasts, we stably expressed the *QIL1* open reading frame (NM_205767.2) tagged on its C-terminus with an HA-FLAG tag using a lentivirus in fibroblasts from patients 1 and 2, and examined mitochondrial structures by electron microscopy after at least 7 days in culture (*Figure 4A–B*). The level of QIL1-HA-FLAG at ~15 kDa was comparable to that of endogenous QIL1 in cells stably expressing ectopic QIL1 based on immunoblotting with anti-QIL1. As noted previously (*Guarani et al., 2015*), ectopic expression of QIL1-HA-FLAG led to a reduction in the abundance of endogenous QIL1 consistent with homeostatic control (*Figure 4B*). Importantly, QIL1 expression converted abnormal mitochondria in patient cells into tubular structures with intact cristae junctions in 70–80% of cells as assessed by electron microscopy (*Figure 4A–B*). We found that QIL1 overexpression resulted in altered cristae morphology in ~30% control neonatal fibroblasts, with mitochondria displaying hyper-branched cristae, which were also termed abnormal (*Figure 4A*, data not shown). Since cells express varying levels of ectopic QIL1, higher QIL1 levels in a subset of cells could be associated with hyper-branched cristae. Moreover, QIL1 expressing cells displayed increased levels of MIC10 and MIC60, consistent with rescue of the biochemical defects in MICOS complex assembly (*Figure 4B*).

## Quantitative proteomics demonstrates loss of MICOS assembly in patient fibroblasts

We previously demonstrated that QIL1 depletion leads to the formation of a MIC60-MIC19-MIC25 sub-complex and failure of MIC10, MIC26 and MIC27 to incorporate into the MICOS complex, resulting in defective MICOS maturation (*Guarani et al., 2015*). To examine the status of MICOS complex assembly in patient fibroblasts, we initially employed blue-native gel electrophoresis. As expected, control fibroblasts contained a major MICOS complex at ~700 kDa as assessed using anti-MIC10 or anti-MIC60 antibodies (*Figure 5A–B*). However, in patient fibroblasts lacking

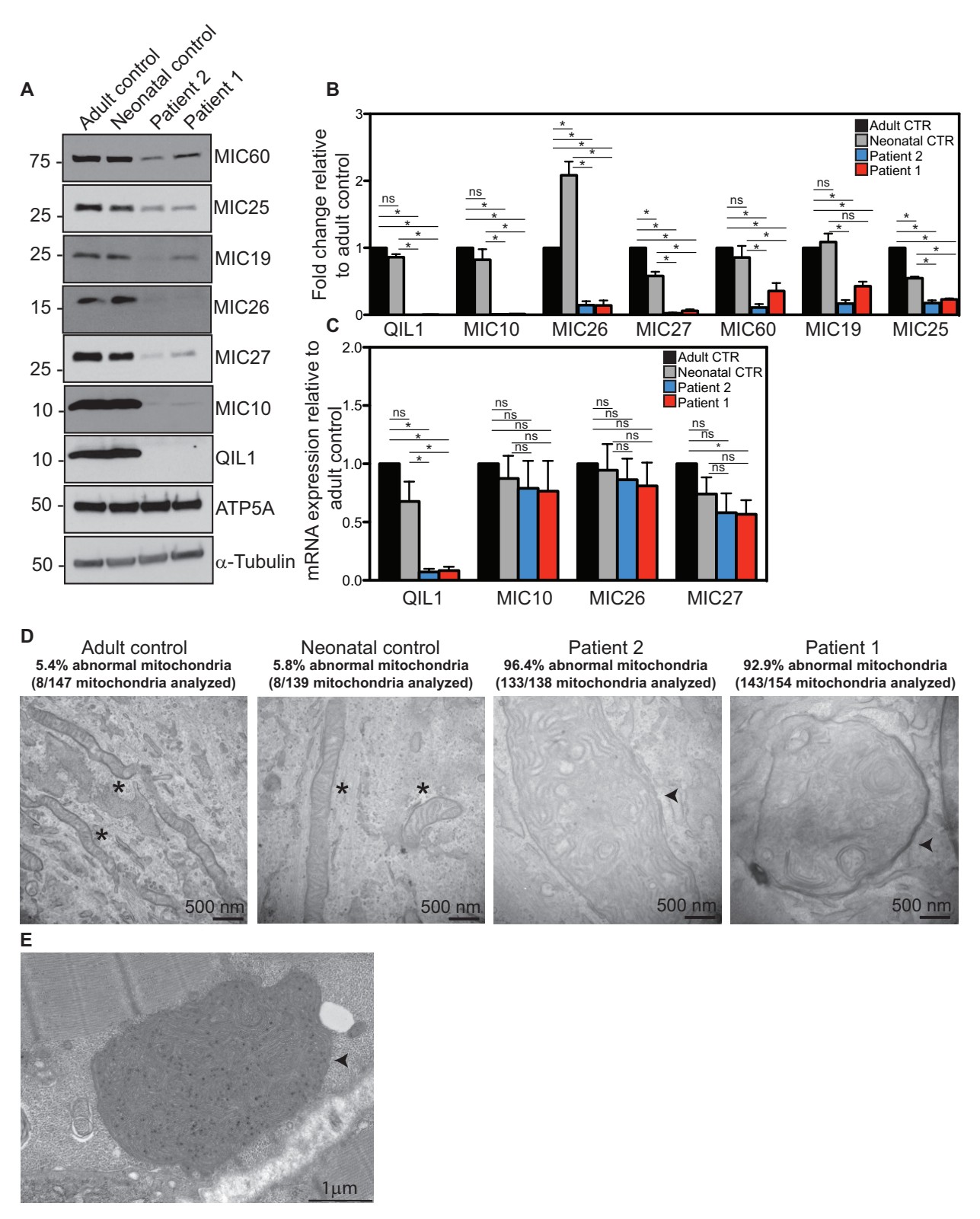

**Figure 3.** Reduced MICOS subunit abundance and cristae morphology defects in QIL1-deficient patients' fibroblasts. (A) Immunoblot analysis of QIL1 and various MICOS subunits in control adult skin fibroblasts, control neonatal skin fibroblasts and skin fibroblasts obtained from patients 1 and 2. Anti-ATP5A and α-Tubulin were used as loading controls. (B) Densitometry analysis was performed using ImageJ. Values were normalized to ATP5A. (C) qPCR analysis. Expression levels were normalized to Tubulin. (D) Electron microscopy analysis of control adult skin fibroblasts, control neonatal skin

*Figure 3 continued on next page*

*Figure 3 continued*

fibroblasts and skin fibroblasts from patients 1 and 2 showing enlarged mitochondria with cristae membrane swirls and proliferation of inner membranes in cells from both patients, as compared to normal mitochondria in control cells; some patients' mitochondria contain electron dense inclusions. Morphologically abnormal mitochondria are indicated by the arrowhead. Mitochondria with cristae junctions of normal morphology are indicated with an asterisk. Quantification of abnormal mitochondria based on analysis of the indicated number of mitochondria by electron microscopy is shown. (E) Electron microscopy analysis of skeletal muscle biopsy from patient 1 showing large round mitochondria (arrowhead), which can sometimes reach the size of two sarcomeres. These large mitochondria show an important proliferation of membranes; some mitochondria contain inclusions in the form of electron dense dots. For panels B and C, asterisks represent p values<0.05. Error bars (± SEM) show the mean of 3 or 4 biological replicates.

QIL1, MIC10 and MIC60 were lost from the ~700 kDa complex, with MIC60 found a lower abundance complex at ~500 kDa (*Figure 5A,B,D*). In contrast, ATP5A complexes representing Complex V of the respiratory chain were unaffected in patient samples (*Figure 5C*).

In order to examine MICOS complex assembly more systematically, we performed quantitative proteomics on native gel purified samples after stable isotope labeling by amino acids in culture (SILAC) (*Figure 5E*). Control fibroblasts were grown in light media and patient 2 fibroblasts were grown in heavy media (Lys-8), and cells mixed 1:1 prior to purification of mitochondria. Mitochondrial protein complexes were then subjected to native gel electrophoresis to separate complexes based on mass, and gel slices from ~66 kDa to >1200 kDa subjected to mass spectrometry (*Figure 5E*). We observed a marked reduction of all detected MICOS subunits at ~700 kDa (*Figure 5F*; H:L ratio <1, green) in patient 2 fibroblasts. Thus, the abundance of the mature heterooligomeric complex (*Harner et al., 2011*; *Ott et al., 2012*) is greatly reduced. Concomitantly, we observed an accumulation of MIC19 and MIC60 in a smaller ~500 kDa sub-complex in patient fibroblasts relative to control fibroblasts (*Figure 5F*; H:L ratio >1, red). These results are consistent with previous proteomic studies of QIL1-depleted cells (*Guarani et al., 2015*) and indicate that loss of QIL1 in patients leads to an overall reduction in mature MICOS and an increase in the abundance of a sub-complex containing MIC19 and MIC60.

## QIL1 expression in patient fibroblasts rescues MICOS assembly

In order to examine whether the absence of QIL1 was critical to the loss of the ~700 kDa MICOS complex, we examined MICOS using blue-native gel analysis of patient fibroblasts rescued with QIL1-HA-FLAG (*Figure 5G–I*). This resulted in the rescue of MICOS complex assembly as assessed using anti-MIC10 on blue-native gels (*Figure 5G*) as well as rescue of MIC10 levels (*Figure 5I*). ATP synthase as monitored by anti-ATP5A in blue-native gels and ATP5A protein levels were used as controls (*Figure 5H,I*).

## Absence of efficient rescue of MICOS assembly and cristae structure in patient fibroblasts ectopically expressing other MICOS subunits

The phenotypes displayed by cells lacking QIL1 suggest a unique function for this protein in MICOS complex integrity. To explore this question further, we individually and stably expressed C-terminally HA-FLAG tagged QIL1, MIC60, MIC19, MIC27, and MIC10 using lentiviruses in fibroblasts from patient 2, and examined mitochondrial morphology by electron microscopy (*Figure 5—figure supplement 1A–K*) and MICOS assembly by blue-native gel analysis (*Figure 5—figure supplement 1L–N*). QIL1-HA-FLAG rescued not only MIC60 and MIC10 (*Figures 4B*, *5I*, *Figure 5—figure supplement 1N*) but also MIC19 and MIC27 levels (*Figure 5—figure supplement 1N*), consistent with formation of MICOS. Expression of MIC60 and MIC19 failed to rescue QIL1-deficiency by electron microscopy (*Figure 5—figure supplement 1F,G*) or by blue-native gel analysis of MIC10 assembly into MICOS (*Figure 5—figure supplement 1L*). In the case of MIC27 and MIC10 overexpression, we observed very low abundance MIC10 at the position of the MICOS complex in blue-native gel analysis, as well as additional low abundance complexes migrating at ~800 kDa and at much lower mass (~100 kDa) (*Figure 5—figure supplement 1L*), suggesting an ability of overexpression of these proteins (*Figure 5—figure supplement 1N*) to support low efficiency or possibly aberrant or alternative assembly. Interestingly, ~30% of individual mitochondria in QIL1-defective fibroblasts overexpressing MIC27 and MIC10 displayed mitochondria with evidence of cristae-like structures (*Figure 5—figure*

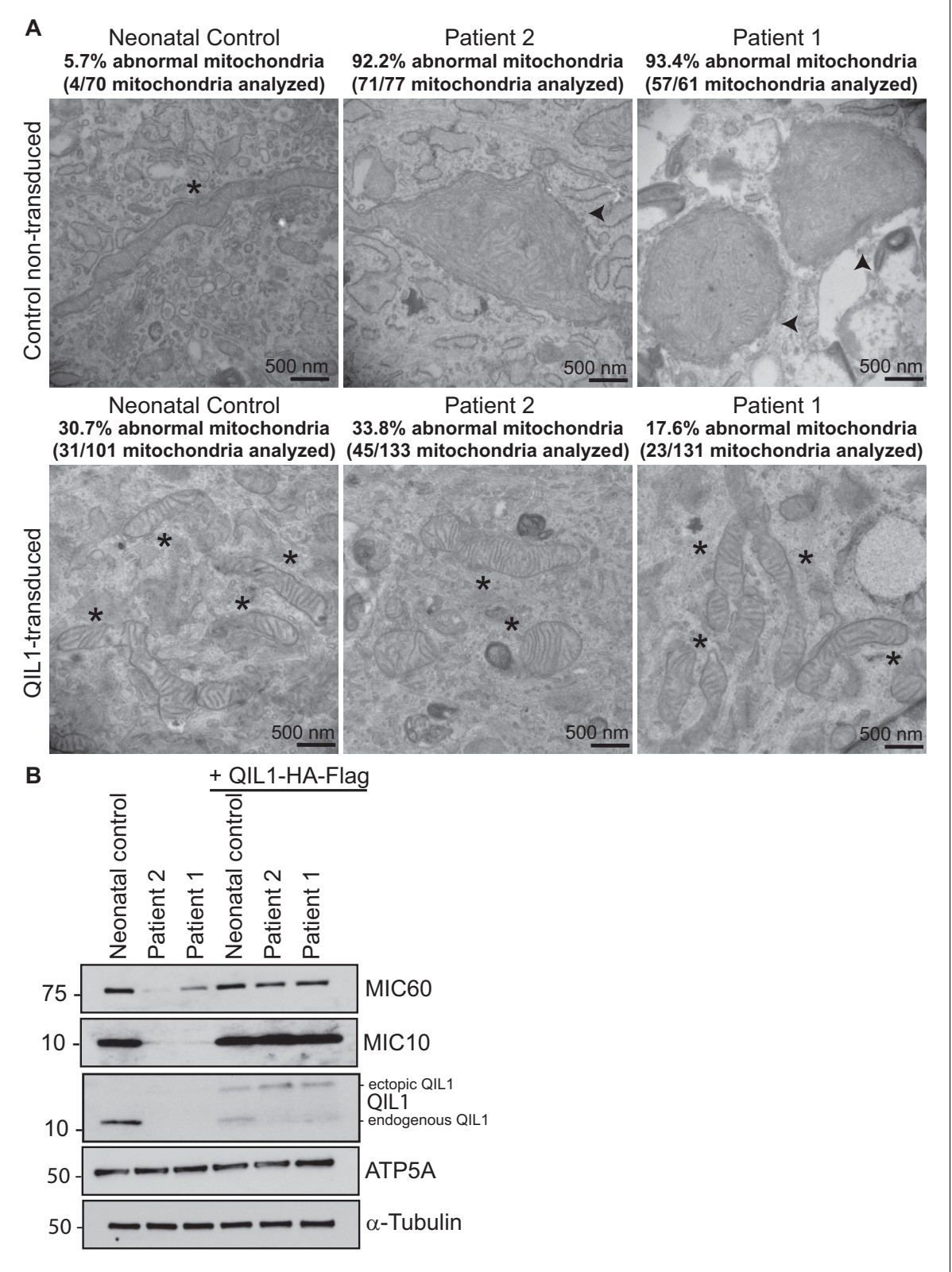

**Figure 4.** Ectopic QIL1 expression rescues mitochondrial morphological defects in patients' fibroblasts. (**A**) Electron micrographs of control neonatal skin fibroblasts and skin fibroblasts from patients 1 and 2 showing the rescue of mitochondria cristae morphology and shape upon ectopic QIL1-HA-Flag expression. Morphologically abnormal mitochondria are indicated by the arrowhead. Mitochondria with cristae junctions of normal morphology are indicated with an asterisk. Quantification of abnormal mitochondria based on analysis of the indicated number of mitochondria by electron

*Figure 4 continued on next page*

*Figure 4 continued*

microscopy is shown. (B) Immunoblot analysis of MICOS subunits MIC10 and MIC60 in control versus patients' skin fibroblasts with or without overexpression of C-terminally HA-Flag tagged QIL1 demonstrating the rescue of the abundance of MICOS subunits upon ectopic QIL1 expression.

supplement 1H,J), while the majority of mitochondria in these cells contained swirl like inner membrane structures typical of disruption of MICOS (*Figure 5—figure supplement 1I,K*). These data suggest a weak ability of MIC27 and MIC10 overexpression to support cristae formation independently of QIL1 (see Discussion).

## Discussion

We report herein QIL1 deficiency in 2 siblings who exhibited early onset severe mitochondrial disease. QIL1 deficiency was associated with transient neonatal deterioration with hypoglycemia, hyperlactacidemia and liver disease. In the first months of life, acute decompensations occurred with liver disease and hyperlactacidemia. This was associated with neurological disease, including developmental delay with cerebellar atrophy, eye disease with optic atrophy and sensorineural deafness. Liver disease also included liver nodules with high α-fetoprotein as observed in other mitochondrial disease such as *TRMU* gene mutations (*Gaignard et al., 2013*).

Liver associated with central nervous system dysfunction is often involved in mitochondrial disease especially those involving defects of mitochondrial DNA maintenance caused by nuclear gene mutations *i.e.* mitochondrial DNA depletion syndromes (resulting from *POLG, DGUOK* or *MPV17* mutations for instance (*Al-Hussaini et al., 2014*; *Naviaux and Nguyen, 2004*). QIL1 deficiency in patient fibroblasts did not affect mitochondrial DNA abundance, but resulted in severely disorganized mitochondrial cristae. In principle, the deletion of exon 2, which encoded the single-pass transmembrane domain apparently involved in QIL1 tethering to the MIM (*Figure 2*), would be expected to greatly reduce the efficiency of MICOS assembly (*Guarani et al., 2015*; *Zerbes et al., 2016*). Given the presence of a premature stop codon in the mRNA for the QIL1 mutant, it is likely that the mRNA is degraded through non-sense mediated decay. We also detected generalized mitochondrial respiratory chain deficiencies in skeletal muscle from patient 1 consistent with a reduced mitochondrial respiration previously reported upon QIL1 RNAi mediated knockdown in human cell lines (*Guarani et al., 2015*). However, we detected only a mild complex IV (cytochrome *c* oxidase) deficiency in skin fibroblasts grown in culture. Interestingly, contrary to what was observed in RNAi-induced QIL1 depletion in HeLa cells (*Guarani et al., 2015*), there was no effect of QIL1 deficiency on fibroblasts' basal respiration in agreement with the limited defect affecting complex IV and the normal respiratory chain enzyme activities in these cells. Differences in mitochondrial activities observed between muscle biopsies and skin fibroblasts from patients are commonly observed in mitochondrial disorders (*Haas et al., 2008*; *Thorburn and Smeitink, 2001*; *van den Heuvel et al., 2004*). It has been reported that nearly 50% of the children with a respiratory chain defect in muscle display normal enzyme activities in cultured skin fibroblasts (*Thorburn, 2000*) and respiratory chain defects are often not expressed in fibroblasts even in some well-recognized oxidative phosphorylation deficiencies (*Robinson et al., 1990*).

Nevertheless, patients' fibroblasts exhibited high sensitivity to glucose deprivation, suggesting an inability to utilize the alternative mitochondrial energetic route to cope with glucose deprivation, in line with previous observations from other mitochondrial diseases (*Robinson et al., 1992*). Interestingly, an increase in cristae density has been reported upon general nutrient or glucose deprivation (*Gomes et al., 2011*; *Rossignol et al., 2004*), indicating an important link between mitochondrial architecture and nutrient utilization. It has been proposed that mitochondrial cristae architecture may control nutrient utilization via different mechanisms, including electron transport chain function, nutrient import and access (*Stanley et al., 2014*). Thus, differences between mitochondrial respiration defects observed in muscle and fibroblasts may not only reflect the pleiotropic nature of phenotypes of mitochondrial diseases, often eliciting highly tissue-specific effects (*Vafai and Mootha, 2012*), but also be related to cell-specific metabolic requirements and different mitochondrial pathways. Alternatively, skin fibroblasts may have possibly developed adaptive mechanisms over time to permit normal basal respiration under culture conditions. Nevertheless, *QIL1* mutation in patient

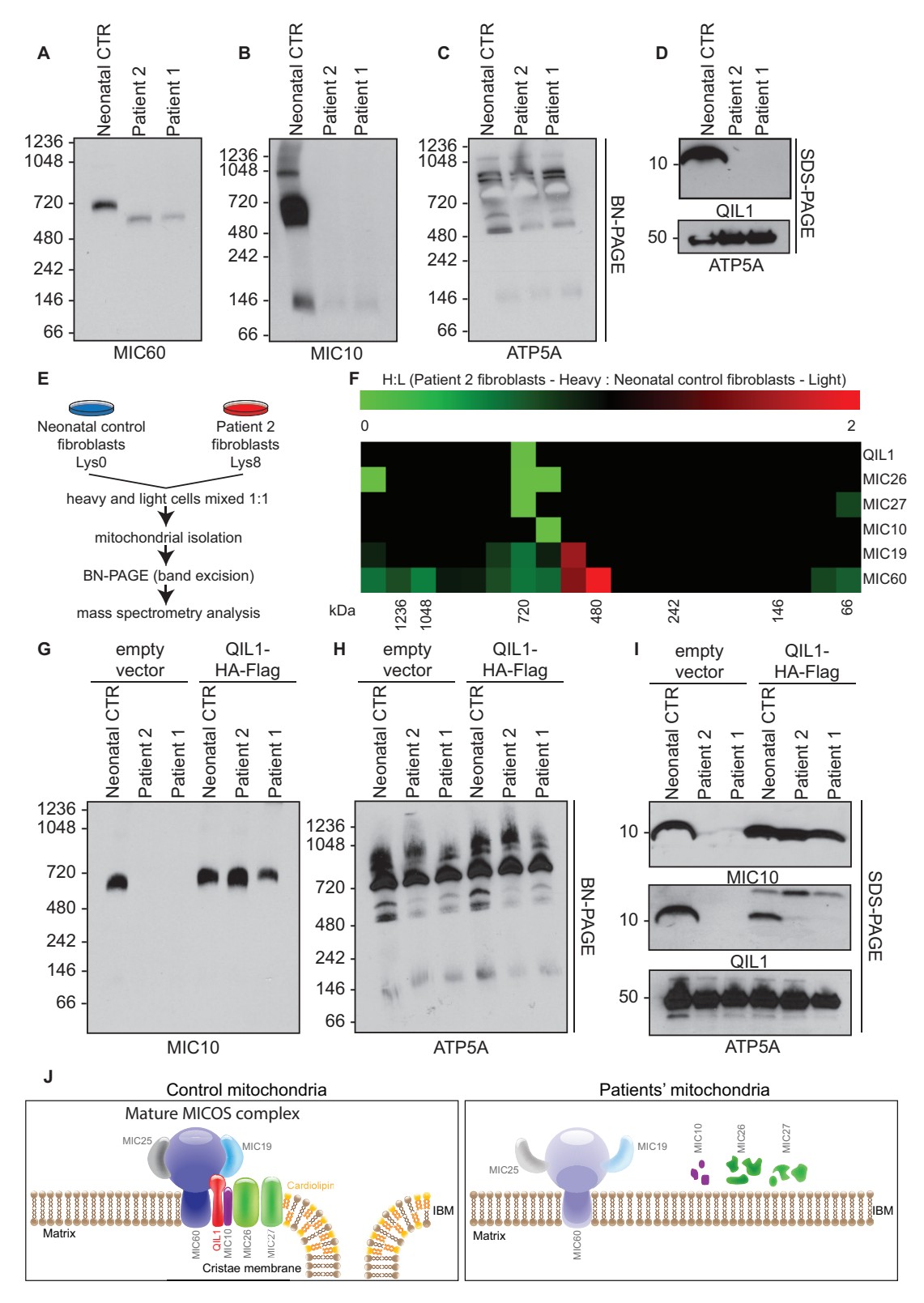

**Figure 5.** Patients' mitochondria display MICOS assembly defects. (A–C) Blue native electrophoresis followed by immunoblot analysis of MIC60, MIC10 and ATP5A in control neonatal and patients' mitochondria from skin fibroblasts. In Panel **D**, mitochondrial extracts were subjected to SDS-PAGE and immunoblotting with the indicated antibodies. (E,F) Cells from control neonatal fibroblasts were labeled with light lysine (K0) while fibroblasts obtained from patient 2 were labelled with heavy lysine (K8). Cells were mixed at a 1:1 ratio and mitochondria were subsequently isolated, lysed in 1% digitonin

*Figure 5 continued on next page*

*Figure 5 continued*

and native protein complexes were separated in a blue native gel. Gel bands from >1 MDa to ~66 kDa were excised and subjected to mass spectrometry analysis. Heavy: Light ratios were calculated for the sum of all peptides quantified for each MICOS subunit and represented in a heatmap where ratios below 1 are represented in green and ratios above 1 are represented in red. (G,H) Blue-native gel analysis of MIC10 and ATP5A containing complexes from neonatal control fibroblasts or fibroblasts from patients 1 and 2 with or without rescue by stable expression of QIL1-HA-FLAG. Panel G, anti-MIC10. Panel H, anti-ATP5A. (I) Mitochondrial lysates from panels **G–H** were examined by SDS-PAGE and immunoblotting using the indicated antibodies. (**J**) Schematic representation of the effect of QIL1 loss on MICOS assembly in patients' fibroblasts. In the mature MICOS complex in control cells, multiple transmembrane components of MICOS (MIC60, MIC10, MIC26, MIC27 and QIL1) associate with MIC25 and MIC19 to promote the formation of cristae membrane structures. In the mitochondria of patient cells lacking QIL1, the abundance of MIC10, MIC26, and MIC27 is greatly reduced, leading to loss of the MICOS complex and the absence of normal cristae structures within mitochondria. Blue-native gel analysis of MICOS from QIL1-deficient cells revealed a ~500 kDa complex containing reduced levels of MIC60 and MIC19, which appears to be unable to maintain cristae structure morphology within mitochondria.

The following figure supplement is available for figure 5:

**Figure supplement 1.** Analysis of QIL1-mutant fibroblasts ectopically expressing MICOS complex subunits.

---

fibroblasts had a major impact on the abundance of other MICOS subunits, on MICOS assembly, and on the mitochondrial ultrastructure. A proposed model is shown in *Figure 5J*, wherein loss of QIL1 leads to loss of cristae structure and reduced levels of MIC60, MIC19, MIC25, MIC10, MIC26, and MIC27, with MIC10, MIC26 and MIC27 being affected to the largest extent (*Figure 3A-B*, *Figure 5I* and *Figure 5—figure supplement 1N*). Residual MIC60 and MIC19 could be found in an ~500 kDa complex in blue-native gels using quantitative proteomics (*Figure 5F,J*), suggesting that this sub-complex of MICOS does not require QIL1 for assembly (*Guarani et al., 2015*). Interestingly, overexpression of MIC10 and MIC27 in QIL1-defective fibroblasts resulted in a fraction of mitochondria containing cristae like structures, with very low levels of MICOS complex assembly observed by blue-native gels. The ability of MIC10 and MIC27 overexpression to promote cristae formation with low efficiency may relate to the ability of MIC10 to form large oligomers and promote membrane curvature independently of other MICOS subunits (*Barbot et al., 2015*; *Bohnert et al., 2015*) and MIC27 to bind cardiolipin which also promotes curvature (*Weber et al., 2013*). Interestingly, we observed that MIC27 expression increased MIC10 levels, and it is possible that the ability of MIC27 to support complex formation with low efficiency may reflect stabilization of MIC10 (*Figure 5—figure supplement 1L,N*). These findings are in line with previous observations by Koob et al who reported that MIC27 expression positively correlated with the levels of MIC10, and proposed that MIC26 and MIC27 modulate MIC10 levels (*Koob et al., 2015*). Zerbes at al also reported that MIC27 promotes the stability of the MIC10 oligomers (*Zerbes et al., 2016*). Further studies are needed to address the underlying mechanism. Additionally, we note that fibroblasts from patients have highly enlarged mitochondrial structures, suggesting loss of appropriate mitochondrial fission-fusion cycles, and the absence of mitophagic destruction of damaged mitochondria. Similar defects in fusion-fission cycles and development of swollen mitochondria has been observed in cells lacking the MICOS subunit MIC60, suggesting that the observed phenotype is a general property of disruption of cristae structures (*Li et al., 2016*).

An additional interesting feature of both patients examined here is persistent urinary excretion of 3-methylglutaconic acid that might also be ascribed to mitochondrial cristae disassembly (*Supplementary file 1A*). 3-methylglutaconic aciduria is observed in other mitochondrial diseases, but most specifically in a new category of nuclear-encoded mitochondrial-related diseases, namely genetic defects of phospholipid biosynthesis, remodeling and metabolism (*Garcia-Cazorla et al., 2015*; *Lu and Claypool, 2015*; *Mayr, 2015*; *Wortmann et al., 2013*, *2015*).

In conclusion, this report demonstrates that primary defects in mitochondrial inner membrane cristae architecture as observed with QIL1 deficiency causes severe mitochondrial disease in human clinically indistinguishable from primary oxidative phosphorylation respiratory chain disease. This is also the first report of early onset fatal mitochondrial encephalopathy in patients with mutations in a core MICOS complex subunit. CHCHD10 haploinsufficiency, found in a family with neurodegenerative disorders and mitochondrial respiratory chain defects also demonstrates impaired MICOS complex stability, cristae formation and mitochondrial genome organization (*Genin et al., 2015*).

Noticeably, alterations of different MICOS subunit protein levels and amino acid substitutions have been associated with several conditions, such as diabetic cardiomyopathy, neurodegeneration and cancer in cellular and animal model systems (*Zerbes et al., 2012*). Taken together, our results suggest that QIL1 and other MICOS subunits represent potential candidates for mutations in patients with unexplained early onset neurological deterioration with optic and cerebellar atrophy progressing over time combined with liver disease and 3-methylglutaconic aciduria.

## Materials and methods

### Brain MRI
Each subject underwent 1.5T brain MRI (Philips Medical Systems, Best, The Netherlands). Three sequences (T1 weighted images, T2, and FLAIR) were scanned by each MRI.

### Cell culture
Skin fibroblast cells were grown in Dulbecco's modified Eagle's medium (DMEM) with Glutamax and 4.5% Glucose, supplemented with 10% fetal calf serum (FBS), 200 µM Uridine and 2.5 mM Pyruvate and maintained in a 5% CO2 incubator at 37°C. Patients' fibroblasts were obtained from skin biopsies of patients and signed informed consent for skin biopsy, fibroblasts storage, enzymatic and molecular analyses was obtained from the parents. Cells were authenticated through an in-house database system from the Biochemistry Department of Bicêtre Hospital (Le Kremlin-Bicêtre, France) where the cells were also stored in liquid nitrogen. Fibroblasts tested negative for mycoplasma. Control fibroblasts were kindly provided by Robin Reed's laboratory (Harvard Medical School) (*Yamazaki et al., 2012*) or obtained from Invitrogen (human dermal fibroblasts; neonatal, Invitrogen [C0045C]). Cells were cultivated in the presence and absence of glucose as reported elsewhere (*Schiff et al., 2011*). On day zero when the experiment was initiated, cells were at 50% confluence. After 4 or 7 days in culture in the presence or absence of glucose, cell density (cell number *per* cm$^2$) was estimated from representative photographs taken using phase-contrast an LSM 5 Exciter optic microscope (Zeiss, Marly le Roi, France) (x40) by counting non-confluent adherent cells in three identical areas of T25 flasks for each condition.

### Antibodies
Antibodies used in this work include: $\alpha$-QIL1 (Sigma-Aldrich, St. Louis, MO; SAB1102836), $\alpha$-MINOS1 (MIC10) (Aviva, San Diego, CA; ARP44801-P050), $\alpha$-CHCHD3 (MIC19) (Aviva; ARP57040-P050), $\alpha$-CHCHD6 (MIC25) (Proteintech, Rosemont, IL; 20639-1-AP), $\alpha$-IMMT (MIC60) (Abcam, Cambridge, MA; ab110329), $\alpha$-APOOL (MIC27) (Aviva ; OAAF03292), $\alpha$-APOO (MIC26) (Novus Bio, Littleton, CO; NBP1-28870), $\alpha$-ATP5A (Abcam; ab14748) and $\alpha$-$\alpha$-Tubulin (Cell Signaling Technologies, Danvers MA; #2125).

### Immunoblot analysis, RNA extraction, reverse transcription and qPCR
These procedures were performed as previously described (*Guarani et al., 2015*).

### Electron microscopy analysis
Control adult skin fibroblasts, control neonatal skin fibroblasts or skin fibroblasts obtained from patients 1 and 2 were cultured on Aclar coverslips, fixed with 1.25% paraformaldehyde, 2.5% glutaraldehyde, 0.03% picric acid followed by osmication and uranyl acetate staining, dehydration in alcohols and embedded in Taab 812 Resin (Marivac Ltd, Nova Scotia, Canada). Sections were cut with Leica ultracut microtome, picked up on formvar/carbon coated copper slot grids, stained with 0.2% Lead Citrate, and imaged under the Phillips Tecnai BioTwin Spirit transmission electron microscope. Muscle electron microscopy studies were performed as previously described (*Malfatti et al., 2014*).

### Quantitative proteomics of MICOS assembly
Mitochondrial proteomics was performed as previously described (*Guarani et al., 2015*). Neonatal control skin fibroblasts or patient 2 skin fibroblasts were grown in light (K0) or heavy media (K8) and an equal number of cells mixed, prior to purification of mitochondria. Mitochondria were lysed with 1% Digitonin and protein complexes fractionated by blue native-polyacrylamide gel electrophoresis

(BN-PAGE). Gel bands were excised and proteins subjected to reduction, alkylation and trypsinization. Tryptic peptides were analyzed using a Q Exactive mass spectrometer (Thermo Fisher Scientific, San Jose, CA) coupled with a Famos Autosampler (LC Packings) and an Accela600 liquid chromatography (LC) pump (Thermo Fisher Scientific). Peptides were separated on a 100 μm inner diameter microcapillary column packed with ~0.25 cm of Magic C4 resin (5 μm, 100 Å, Michrom Bioresources, Auburn, CA) followed by ~18 cm of Accucore C18 resin (2.6 μm, 150 Å, Thermo Fisher Scientific). For each analysis, we loaded ~1 μg onto the column. Peptides were separated using a 90 gradient of 5 to 28% acetonitrile in 0.125% formic acid with a flow rate of ~300 nL/min. The scan sequence began with an Orbitrap MS1 spectrum with the following parameters: resolution 70,000, scan range 300−1500 Th, automatic gain control (AGC) target $1 \times 10^6$, maximum injection time 250 ms, and centroid spectrum data type. We selected the top twenty precursors for MS2 analysis which consisted of HCD high-energy collision dissociation with the following parameters: resolution 17,500, AGC $1 \times 10^5$, maximum injection time 60 ms, isolation window 2 Th, normalized collision energy (NCE) 25, and centroid spectrum data type. The underfill ratio was set at 9%, which corresponds to a $1.5 \times 10^5$ intensity threshold. In addition, unassigned and singly charged species were excluded from MS2 analysis and dynamic exclusion was set to automatic.

## Mitochondrial respiratory chain activity measurement

Fibroblasts were trypsinized and centrifuged 5 min × 1500 g. The supernatant was discarded and the pellet washed (5 min × 1500 *g*) with 1 mL PBS. The majority of the fresh pellet was used for oxygen consumption measurement (*El-Khoury et al., 2013*; *Rustin et al., 1994*). A small aliquot of the pellet was deep-frozen in 20–40 μL PBS solution and subsequently thawed using 1 mL of ice-cold solution consisting of 0.25 M sucrose, 20 mM Tris (pH 7.2), 2 mM EGTA, 40 mM KCl and 1 mg/mL BSA, 0.004% digitonin (w/v), and 10% Percoll (v/v) (medium A). After 7 min incubation at ice temperature, cells were centrifuged (5 min × 2300 *g*), the supernatant discarded, and the pellet washed (5 min × 2300 *g*) with 1 mL of medium A devoid of digitonin and Percoll. The final pellet was re-suspended in 20–30 μL of this medium and used for spectrophotometrical enzyme assays.

Respiratory chain enzyme activities were spectrophotometrically measured using a Cary 50 UV–visible spectrophotometer (Varian Inc, Les Ulis, France) (*Bénit et al., 2006*; *Rustin et al., 1994*).

## Oxygen uptake and mitochondrial substrate oxidation

Oxygen uptake and substrate oxidation were measured using a microoptode consisting in an optic fiber equipped with an oxygen sensitive fluorescent terminal sensor (FireSting O2; Bionef, Paris, France) as reported previously (*El-Khoury et al., 2013*).

All chemicals were of the purest grade available from Sigma-Aldrich (St Quentin Fallavier, France). Protein concentration was measured according to Bradford.

Respiratory chain enzyme activities were determined in skeletal muscle from patient 1 as previously reported (*Medja et al., 2009*).

## Whole exome sequencing

Patients 1 and 2 DNA extracted from fibroblasts was sequenced by the IGBMC Microarray and Sequencing platform (Strasbourg, France). Exon-capture was performed using the SureSelect XT2 Human all exon V5 enrichment System (Agilent, Santa Clara, CA) followed by Hiseq 2500 sequencing (Illumina Inc) according to the manufacturer's protocol. Image analysis and base calling were performed using the Real-Time Analysis from Illumina. DNA sequences were aligned to the reference genome Hg19 using BWA v0.6.1 (*Li and Durbin, 2009*). Aligned data were then refined with the use of Picard v1.68 (http://picard.sourceforge.net/) to flag duplicate reads, GATK v2.5.2 (*DePristo et al., 2011*) to perform local realignments and recalibrate base qualities, and Samtools v0.1.18 (*Li and Durbin, 2009*) to filter out multi-mapped reads. Variant calling was done by combining the results of 4 variant callers i.e GATK UnifiedGenotyper, GATK HaplotypeCaller, Samtools v0.1.18 mpileup and Samtools 0.1.7 pileup. Variant quality scores were recalibrated using GATK. Variants were annotated using SnpEff v2.0.5, GATK v2.5.2 and SnpSift v3.3c (*Cingolani et al., 2012a*, *2012b*). Finally, VaRank (*Geoffroy et al., 2015*) was used to rank discovered variants, which were filtered on homozygous state in the two affected patients. Variants

identified by exome sequencing were confirmed by Sanger sequencing (primers are available on *Supplementary file 2*).

## Reverse transcription (RT) reactions and cDNA sequencing

These procedures were performed using the QIAGEN OneStep RT-PCR Kit (Qiagen, Hilden, Germany) following the manufacturer's instructions and using 0.5 µl of RNA at 300 ng/µl (primers are available on *Supplementary file 2*).

## Mitochondrial DNA content

Mitochondrial DNA content was determined in skeletal muscle as described previously (*Kim et al., 2008*).

## Acknowledgements

This work was supported by E-rare GENOMIT, 'Ouvrir les Yeux' and AMMi (Association contre les Maladies Mitochondriales) grants to PR, DC, PB and MS, and NIH grants GM095567 and NS083524 to JWH. CJ received funding from the *Fondation Maladies Rares*. We thank Maria Ericsson and Elizabeth Benecchi at the Conventional Electron Microscopy facility at Harvard Medical School for outstanding assistance with electron microscopy. We thank the IGBMC Microarray and Sequencing platform, member of the 'France Génomique' consortium (ANR-10-INBS-0009), for WES of patients.

## Additional information

### Competing interests

JWH: Reviewing Editor, *eLife*. The other authors declare that no competing interests exist.

### Funding

| Funder | Grant reference number | Author |
| --- | --- | --- |
| Fondation Maladies Rares | | Claude Jardel |
| Ouvrir les Yeux | | Dominique Chrétien<br>Paule Bénit<br>Pierre Rustin<br>Manuel Schiff |
| E-rare GENOMIT | | Dominique Chrétien<br>Paule Bénit<br>Pierre Rustin<br>Manuel Schiff |
| Association contre les Maladies Mitochondriales | | Dominique Chrétien<br>Paule Bénit<br>Pierre Rustin<br>Manuel Schiff |
| National Institutes of Health | GM095567 | J Wade Harper |
| National Institutes of Health | NS083524 | J Wade Harper |

The funders had no role in study design, data collection and interpretation, or the decision to submit the work for publication.

### Author contributions

VG, JWH, MS, Conception and design, Acquisition of data, Analysis and interpretation of data, Drafting or revising the article; CJ, DC, AL, CL, EL, AB, AI, J-FB, ID, MG, ESG, PG, AS, ME-B, NBR, PR, HOdB, JAP, Conception and design, Acquisition of data, Analysis and interpretation of data; PB, Acquisition of data, Analysis and interpretation of data

### Author ORCIDs

J Wade Harper, http://orcid.org/0000-0002-6944-7236

### Ethics

Human subjects: The ethics committee of Robert Debré University Hospital (APHP, 75019 Paris, France) approved the study of human fibroblasts. All procedures conformed to ethical standards. Informed written consent was obtained from parents.

# Additional files

### Supplementary files

• Supplementary file 1. Clinical and laboratory findings and respiratory chain activities in control and QIL1-defective patients. (A) Clinical and laboratory findings from patients 1 and 2. Nd: non determined; H: hours of life; mo: months of life. 3MGCA: 3-methylglutaconic aciduria. (B) Respiratory chain activities in control and QIL1-defective patients' fibroblasts. In fibroblasts, patients' respiratory chain complexes activities were in control ranges. Complex IV activity normalized to citrate synthase was however more than 3 standard deviations lower than control, indicative a partial defect in complex IV. Activities are expressed in nmol/min/mg protein. (C) Respiratory chain activities in control and QIL1-defective patients' skeletal muscle for patient 1. In muscle, all respiratory chain complexes (I-IV) activities normalized to citrate synthase were defective. Activities are expressed in nmol/min/mg protein.

• Supplementary file 2. Primer sequences for QIL1 genomic DNA (gDNA) and complementary DNA (cDNA). Tables show QIL1 primers used for PCR amplification.

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
