## [Decision Letter]

Thank you for submitting your article "QIL1 variants cause MICOS disassembly and early onset fatal mitochondrial encephalopathy with liver disease" for consideration by *eLife*. Your article has been reviewed by three peer reviewers, and the evaluation has been overseen by Randy Schekman as the Reviewing Editor and Senior Editor. The following individuals involved in review of your submission have agreed to reveal their identity: Toni Barrientos (Reviewer #1); Jared Rutter (Reviewer #3).

The reviewers have discussed the reviews with one another and the Reviewing Editor has drafted this decision to help you prepare a revised submission.

Summary:

The conserved mitochondrial contact site and cristae junction (MICOS) complex functions as a primary determinant of mitochondrial inner membrane structure. Among the many components previously identified, the authors of this manuscript focus on QIL1, a relatively novel component, recently identified by the authors. The relevance of the present submission is that they have now found that null mutations in this gene, identified by exome sequencing, result in a recessive mitochondrial disorder in humans, involving early-onset fatal encephalomyopathy and liver disease. The analysis of patients' fibroblasts have allowed to confirm the central role on QIL1 in MICOS assembly and function.

The authors specifically show that the lack of QIL1 induces degradation of most remaining MICOS subunits, particularly MIC10 and MIC26, and induce the accumulation of a MICOS subcomplex containing MIC19 and MIC60. As expected, reintroducing wild-type QIL1 in patient's fibroblasts restores MICOS assembly and function.

The manuscript is technically and conceptually sound. The data and figures are excellent. The novelty lies on the identification of disease-associated mutations in QIL1 but not in the characterization of QIL1 function, since all details presented here in patient's fibroblasts were previously reported by the authors using QIL1 siRNA experiments.

Recommended revisions:

1) The authors could have tried to gain more mechanistic insight into how essential QIL1 is in MICOS formation by for example, attempt suppression of the deleterious phenotype in QIL1-deficient fibroblasts by overexpression of some of the other MICOS subunits. Or, does QIL1 overexpression induces any changes in MICOS formation? This would help understanding whether QIL1 has a regulatory role in MICOS formation.

2) The biochemical data presented are not terribly convincing in my view. Control values are presented as ranges with no standard deviation, and the single control they chose appears to be right very near the bottom of the control range (for fibroblasts). There may be a CIV defect in the patient fibroblasts, but it is awfully mild. No controls were analyzed for muscle and again only ranges are given.

3) I did not find the pictures of the patient cells growing in glucose vs. galactose particularly helpful. It doesn't look like the numbers are that different from control (ie the cells appear to have proliferated), and I guess they interpret the brightness as a stress response, which is not unreasonable, but I must say I find it a bit surprising based on the intact respiratory chain. Cell counts, some molecular markers would be useful. In my experience some cells with combined respiratory chain defects more severe than those reported here are able to grow on galactose, and such a comparison might provide some useful insight.

4) What is the relevance of the increased level of PCNA in the patient cells? Figure 5 does not have a C in the legend. (One can also see this in 4A).

5) I think that they should show the biochemical phenotype in the rescued patient cells in Figure 6. It is curious to me that they omitted that.

6) In the Abstract, the authors specifically point out a complex IV defect. It seems that these patients do not have a specific defect in complex IV but a more general defect in respiration.

7) Please comment on potential reasons why the respiratory phenotype varies between fibroblasts and muscle biopsy. Could this be related to specific biochemistry related to MICOS?

8) The pedigree in Figure 3 does not use standard arrangement. Please revise. It also indicates (given that it is not done in a standard way) that the parents are consanguineous, which the text claims to not be true.

9) Consider demonstrating that re-expression of QIL1 in patient fibroblasts is capable of rescuing the complex assembly defects described in Figure 6A-C. It would be nice to see that re-expression will restore steady state levels of the fully assembled MICOS complex.

[Editors' note: further revisions were requested prior to acceptance, as described below.]

Thank you for resubmitting your work entitled "QIL1 mutation causes MICOS disassembly and early onset fatal mitochondrial encephalopathy with liver disease" for further consideration at *eLife*. Your revised article has been favorably evaluated by Randy Schekman (Senior editor), a Reviewing editor, and three reviewers.

The manuscript has been improved but there is one remaining issue that needs to be addressed before acceptance. Reviewer #3 asks for clarification of the experiment in Figure 1. In consultation session the other reviewers agreed this should be addressed in a response with a possible revision. I will attempt to expedite a final decision when you respond to this inquiry.

Reviewer #1:

The authors have responded to all previous concerns. I believe the manuscript is now in good shape for publication in *eLife*.

Reviewer #2:

The authors took the comments of all reviewers to heart, with the result that the manuscript is much improved, now with some functional data. I have no further comments and suggest accepting the manuscript as submitted.

Reviewer #3:

In general, I believe that the paper is acceptable for publication at this point. I do have a problem, however, with Figure 1. Either I don't understand the experiment (and I believe it is explained incompletely) or all of the three cell lines grow normally through 4 days of the experiment. only after this point, the patient cells start dying while the controls continue to divide. If that's the case, there is something more complicated happening here. If they can clarify this issue, I am ready for it to be accepted.

---

## [Author Response]

*Recommended revisions:*

*1) The authors could have tried to gain more mechanistic insight into how essential QIL1 is in MICOS formation by for example, attempt suppression of the deleterious phenotype in QIL1-deficient fibroblasts by overexpression of some of the other MICOS subunits. Or, does QIL1 overexpression induces any changes in MICOS formation? This would help understanding whether QIL1 has a regulatory role in MICOS formation.*

We agree with the reviewer’s that understanding QIL1 function is an important endeavor. One suggestion was to overexpress QIL1 and examine the effect on MICOS formation. In our previous paper in *eLife* identifying QIL1, we performed this experiment. Briefly, we found that in cells containing a lentiviral vector ectopically expressing QIL1-HA under CMV control, we were unable to “overexpress” the protein, and in the context of QIL1-HA expression, there was a commensurate reduction in the abundance of the endogenous protein, suggesting a homeostatic mechanism controlling total QIL1 abundance. In the present study, we observed a similar reduction of endogenous QIL1 levels upon expression of the C-terminally tagged version (Figure 4 and Figure 5). Of note, ectopic QIL1 expression did not seem to affect MICOS levels or assembly as evidenced by MIC10 immunoblotting of Blue-native gels (Figure 5) and SDS-PAGE analysis of MIC60 and MIC10 protein levels (Figure 4 and Figure 5).

In order to further address this question, we ectopically expressed MIC60, MIC19, MIC27, MIC10, and QIL1 in patient fibroblasts lacking QIL1 and examined the MICOS complex by blue-native gel using anti-MIC10 for detection, and cristae morphology by electron microscopy (new Figure 5—figure supplement 1). We found that neither MIC60 nor MIC19 rescued MICOS assembly by blue-native electrophoresis (Figure 5—figure supplement 1). In the case of MIC27 and MIC10 overexpression, we detected anti-MIC10 reactivity very weakly at ~700 kDa in the native gel analysis, but we also observed additional forms at ~800 kDa and at much lower mass (~100 kDa) (Figure 5—figure supplement 1). Similarly, by EM, expression of MIC60 or MIC19 did not promote cristae structures. Interestingly, we found that a small fraction (~30%) of mitochondria in cells expressing MIC27 and MIC10 display some level of cristae structures (see Figure 5—figure supplement 1). This was a surprising result but was consistent with the detection of MIC10 positive structures by native PAGE in cells over-expressing MIC27 or MIC10. As noted in the previous version, loss of QIL1 leads to a virtually complete loss of MIC10 protein in mitochondrial extracts (see Figure 5—figure supplement 1). Interestingly, overexpression of MIC27 led to the accumulation of endogenous MIC10 in QIL1 null cells (see Figure 5—figure supplement 1). This suggests that MIC27 may stabilize endogenous MIC10, which may facilitate partial assembly of a MICOS-like complex capable of forming cristae-like structures by electron microscopy in the absence of QIL1. These findings are in line with previous observations by Koob et al. (Koob et al., 2015) who reported that MIC27 expression positively correlated with the levels of MIC10 and suggested that MIC26 and MIC27 modulate MIC10 levels. Zerbes at al. also reported that MIC27 promotes the stability of the MIC10 oligomers (Zerbes et al., 2016). Moreover, Barbot et al. and Bohnert et al. (Barbot et al., 2015; Bohnert et al., 2015) have shown that MIC10 forms large oligomers and induces membrane curvature independently of other MICOS subunits, which could contribute to the partial normalization of cristae structures observed in patients’ fibroblasts expressing ectopic MIC27 and MIC10. We now comment on this in the Discussion.

We appreciate the reviewer suggesting this experiment.

*2) The biochemical data presented are not terribly convincing in my view. Control values are presented as ranges with no standard deviation, and the single control they chose appears to be right very near the bottom of the control range (for fibroblasts). There may be a CIV defect in the patient fibroblasts, but it is awfully mild. No controls were analyzed for muscle and again only ranges are given.*

Our data showed that muscle biopsies obtained from one of the two QIL1-deficient patients displayed generalized reduction of mitochondrial respiratory chain activities, while only mild activity ratios defects (cytochrome c oxidase to citrate synthase ratio) and normal overall respiration were observed in cultured skin fibroblasts from both patients. This is however a feature commonly observed in mitochondrial disorders. Even though skin biopsies are more commonly performed than muscle biopsies due to the higher invasiveness associated with muscle biopsies, as much as 50% of the children with a respiratory chain defect in muscle display normal enzyme activities in cultured skin fibroblasts (Thorburn, 2000). High respiratory activities can be measured in fibroblasts even in cases of genetically proven mitochondrial diseases (Thorburn and Smeitink, 2001) and overall, enzyme defects observed in muscle are typically more pronounced than in fibroblasts (van den Heuvel et al., 2004). Indeed, respiratory chain defects are not expressed in fibroblasts even in some well-recognized oxidative phosphorylation deficiencies (Robinson et al., 1990). Thus, fibroblasts are considered less reliable than muscle biopsies for the diagnosis of respiratory chain diseases and an absence of defects in fibroblast biochemical analysis is not sufficient to exclude a disorder of mitochondrial metabolism (Haas et al., 2008). The partial defect noticed in these fibroblasts was the incentive to carry out the experiment in the presence or absence of glucose, as respiratory chain defective fibroblasts often rely on glucose-dependent activity to proliferate and survive (Robinson et al., 1992). Indeed, patients’ fibroblasts exhibited high sensitivity to glucose deprivation (Figure 1), suggesting an inability to utilize the alternative mitochondrial energetic route to cope with glucose deprivation. We now comment on this in the Discussion.

Regarding the reviewer’s comment on why no controls were analyzed for muscle and ranges are given instead, this is due to previous observations that have shown that absolute values for activities of mitochondrial enzymes are extremely variable, particularly in skeletal muscle samples from hospitalized individuals (non-suspected of mitochondrial disease) (Rustin et al., 1991). The distribution of these values being in addition nonstandard (possibly distorted by the limited exercise done by part of the hospitalized patients), it would be rather unsafe to use standard deviations (Chretien et al., 1998). This is why in major mitochondrial diagnostic centers in France (where the two QIL1 defective patients reported herein were hospitalized) data are compared to ranges of control activities. Varying values for absolute activities are indeed observed in most if not all types of human tissues (liver, lymphocytes, including cultured skin fibroblasts). Indeed, we have previously reported that the scatter of control absolute respiratory activity values and their overlap with those of patients limits the diagnosis of respiratory chain deficiencies (Rustin et al., 1991). However, respiratory enzyme activities are maintained at a constant ratio in every tissue analyzed (Rustin et al., 1991), which may reflect the need for the maintenance of the appropriate stoichiometry between oxidative phosphorylation complexes for the optimum function of the electron transfer process. These observations prompted us to present enzyme activities as ratios rather than absolute values. Of note, obtaining control muscle tissues covering a given range of ages and appropriate healthy tissues is often a challenge (Thorburn and Smeitink, 2001). Thus, data presented in [Supplementary-material SD1-data] are given either as ranges (for absolute enzyme activities) or as means +/- standard deviations (for ratios of enzyme activities).

A typing error in normal range of CIV has been corrected: 130-195 instead of 130-95. Ranges of respiratory chain activities were assessed in 89 control patients as described (Medja et al., 2009) and not 25 as erroneously written in the original version of paper. Additionally, typing errors in the [Supplementary-material SD1-data] activity ratios have also been corrected. We apologize.

*3) I did not find the pictures of the patient cells growing in glucose vs. galactose particularly helpful. It doesn't look like the numbers are that different from control (ie the cells appear to have proliferated), and I guess they interpret the brightness as a stress response, which is not unreasonable, but I must say I find it a bit surprising based on the intact respiratory chain. Cell counts, some molecular markers would be useful. In my experience some cells with combined respiratory chain defects more severe than those reported here are able to grow on galactose, and such a comparison might provide some useful insight.*

We agree with the reviewer and therefore discarded the uninformative micrograph in order to present a quantification of cell density after culture for 4 or 7 days in the presence or absence of glucose (new Figure 1). The data indicated that patient fibroblasts are sensitive to glucose depletion, and that, even if partial, the unbalanced activity of enzymes evidenced in these fibroblasts is deleterious. Moreover, QIL1 deficiency and the resulting abnormal cristae architecture may contribute to the cells’ inability to grow without glucose in different ways. For example, an increase in cristae density has been reported upon general nutrient or glucose deprivation (Gomes et al., 2011; Rossignol et al., 2004) indicating an important link between mitochondrial architecture and nutrient utilization. It has been proposed that mitochondrial cristae architecture may control nutrient utilization via different mechanisms including electron transport chain function, but also nutrient import and access (Stanley et al., 2014). Thus, additional factors may contribute to the severity of the observed phenotype.

*4) What is the relevance of the increased level of PCNA in the patient cells? Figure 5 does not have a C in the legend. (One can also see this in 4A).*

We thank the reviewer for noting the inconsistency between the legend and the figure. This has been repaired.

With respect to PCNA, we think that this might have biological significance. For this reason, PCNA may not be the best loading control to use. We have now re-probed the membranes (original Figure 4, Figure 5 updated to Figure 3 and Figure 4) with a different control blot protein – α-tubulin antibody. The levels of α-tubulin were unchanged across all conditions. We have now replaced these panels with a different control blot protein – α-tubulin – which shows equal loading. We note that ATP5A used as a loading control for mitochondria was also equivalent across all the genotypes and ATP5A is used as a loading control throughout the manuscript.

*5) I think that they should show the biochemical phenotype in the rescued patient cells in Figure 6. It is curious to me that they omitted that.*

We appreciate the reviewer’s comments. In the original Figure 6, we provided evidence that in patients’ fibroblasts, MICOS assembly is defective. Blue native electrophoresis analysis followed by immunoblotting with MIC60 and MIC10 antibodies and SILAC quantitative proteomics revealed the disassembly of the mature heterooligomeric MICOS complex at ~700 kDa with the accumulation of a smaller sub-complex at ~500 kDa containing MIC60 and MIC19, in line with our previous observations with RNAi-mediated QIL1 knockdown (Guarani et al., 2015). The reviewer suggested that we show the biochemical phenotype in the rescued patient cells in Figure 6, now updated to Figure 5. We now demonstrate that QIL1 re-introduction rescues the biochemical phenotypes in multiple ways: 1) strong accumulation of MIC10 in a ~700 kDa MICOS complex by native gel analysis (Figure 5, Figure 5—figure supplement 1), 2) strong recovery of MIC10 and MIC27 levels (Figure 5, Figure 5—figure supplement 1), and partial recovery of MIC60 and MIC19 levels (Figure 5—figure supplement 1). Thus, QIL1 re-introduction rescues the major biochemical phenotypes observed in patients lacking QIL1.

*6) In the Abstract, the authors specifically point out a complex IV defect. It seems that these patients do not have a specific defect in complex IV but a more general defect in respiration.*

We agree with the reviewer that in skeletal muscle there is a general defect in respiration. However, a mild CIV deficiency was observed in skin fibroblasts and mitochondrial respiration was unaffected. We discussed potential reasons for the differences in the degree of respiratory defects observed between fibroblasts and muscle biopsy under point 2 above. We have also modified the Abstract accordingly.

*7) Please comment on potential reasons why the respiratory phenotype varies between fibroblasts and muscle biopsy. Could this be related to specific biochemistry related to MICOS?*

We have discussed potential reasons why the respiratory phenotype varies between fibroblasts and muscle biopsy under point 2 above. Additionally, the reviewer’s raises an interesting possibility that the variation observed between the respiratory phenotypes in fibroblasts and muscle may be due to tissue specific biochemistry related to MICOS. As far as mitochondrial architecture is concerned, QIL1 depletion had a similar impact in fibroblasts and muscle tissue as demonstrated in updated Figure 3. Due to unavailability of tissue samples, we are unable to address how MICOS subunits levels and assembly in patients’ skeletal muscle compare to skin fibroblasts.

*8) The pedigree in Figure 3 does not use standard arrangement. Please revise. It also indicates (given that it is not done in a standard way) that the parents are consanguineous, which the text claims to not be true.*

We apologize and thank the reviewer for noticing this inconsistency. The parents are indeed non-consanguineous and the double bar has been corrected to a single bar in updated Figure 2. The pedigree in Figure 2 now uses standard arrangement.

*9) Consider demonstrating that re-expression of QIL1 in patient fibroblasts is capable of rescuing the complex assembly defects described in Figure 6A-C. It would be nice to see that re-expression will restore steady state levels of the fully assembled MICOS complex.*

We performed this analysis as described under point 5 above.

[Editors' note: further revisions were requested prior to acceptance, as described below.]

*The manuscript has been improved but there is one remaining issue that needs to be addressed before acceptance. Reviewer #3 asks for clarification of the experiment in Figure 1. In consultation session the other reviewers agreed this should be addressed in a response with a possible revision. I will attempt to expedite a final decision when you respond to this inquiry.*

*Reviewer #3:*

*In general, I believe that the paper is acceptable for publication at this point. I do have a problem, however, with Figure 1. Either I don't understand the experiment (and I believe it is explained incompletely) or all of the three cell lines grow normally through 4 days of the experiment. only after this point, the patient cells start dying while the controls continue to divide. If that's the case, there is something more complicated happening here. If they can clarify this issue, I am ready for it to be accepted.*

We appreciate the comments of the reviewer. We apologize for not making this aspect of the paper clearer. In order to address this question, we have:

1) redrawn Figure 1 to make the comparisons clearer (and updated the legend accordingly),

2) rewritten the sentence in the Results section to make it clearer what the conclusion of the experiment is, and

3) improved the methods by adding more details.

The goal of the experiment in Figure 1 was to determine if cells lacking QIL1 expression were able to cope with the metabolic challenge of glucose deprivation, as this phenotype is often observed with cells harboring mitochondrial mutations (Robinson et al., 1992). To address this question, we cultured control and patient fibroblasts in the presence and absence of glucose and measured cell numbers at 4 and 7 days. On day zero when the experiment was initiated, cells were at 50% confluence. Importantly, the cells employed are primary fibroblasts and their proliferation rates are quite slow compared with other commonly used cell types. As can be seen in the presence of glucose in Figure 1, cells double in about 3 days, a feature that is important for interpreting the data but may not have been obvious in the revised version of the paper. The primary conclusion that we make is after 7 days in the absence of glucose, the control cells continue to proliferate while the patient cells do not. This demonstrates what we think is a dramatic sensitivity to the absence of glucose and think this conclusion is clear. We interpret the reviewer’s comments in part to be asking why there isn’t an effect already at day 4. We believe this reflects two major factors. First, the slow cell cycle rates for these cells coupled with the time required to move from cell cycle arrest through apoptosis likely affect the total number of cells remaining at this early time point, making it difficult to observe changes. Second, previous work using patient cells with mutations in mitochondrial systems vary in the point in time that they begin to be sensitive to glucose deprivation (Robinson et al., 1992). This likely reflects the severity of the phenotype as well as complex circuits and possibly reprogramming of metabolic pathways or utilization of alternative energy sources (Stanley et al., 2014), the elucidation of which we feel is beyond the scope of this work. We also note that because these experiments are using primary cells from individuals, there are sometimes differences in growth rates among the individual patients even in the presence of glucose. Overall, the slight variation in cell numbers between the different cell lines likely represents variation at the individual level.

We now state in the text: “Compared with control fibroblasts, patient fibroblasts displayed reduced cell number 7 days after glucose withdrawal (Figure 1), a characteristic often observed in mitochondrial disease and indicative of a limited capacity for mitochondria to respond to the metabolic challenge resulting from glycolysis limitation (Lee et al., 2014; Robinson et al., 1992; van den Heuvel et al., 2004)”. We can explicitly point out the variation in onset to sensitivity if the editors feel it is necessary.